# Benchmarking Attention for Tabular Foundation Models

**Maximilian Schambach** [1]   **Clemens Biehl** [1]   **Sam Thelin** [1]

## Abstract

Tabular in-context learners such as TabPFN rely on alternating row and column attention over 2D sequences of latent embeddings. These attention patterns differ markedly from the one-dimensional case in language models: row attention involves longer sequences while column attention operates on much shorter ones. Furthermore, the strided memory layout of the data makes producing contiguous tensors costly. Yet efficient attention has been studied mostly for one-dimensional sequences, leaving the two-dimensional tabular setting unexplored. To this end, we study the unique characteristics of tabular attention and benchmark different backends – Torch SDPA (efficient and cuDNN) and FlashAttention-2, 3, and 4 – measuring throughput across realistic tabular shapes on an H100 GPU. We find that the optimal choice differs between column and row attention: cuDNN-based SDPA dominates column attention at short-to-medium sequence lengths, while FlashAttention-3 dominates row attention by supporting a more compact stride layout directly, avoiding costly tensor copies.

## 1. Introduction

Tabular data accounts for the majority of data in enterprise machine learning applications (Chui et al., 2018). While gradient-boosted decision trees have long been the dominant paradigm for tabular prediction (Grinsztajn et al., 2022), recent pretrained predictive in-context learners such as TabPFN (Hollmann et al., 2025), TabICL (Qu et al., 2025; 2026), Mitra (Zhang et al., 2025d), or ConTextTab (Spinaci et al., 2025) have demonstrated strong, often state-of-the-art performance on public benchmarks (Erickson et al., 2025). At their core, these models are transformer-based and share a common architectural design: alternating *row attention*

(attending across data samples) and *column attention* (attending across features) (Hollmann et al., 2025), as full cross-cell attention would be prohibitively expensive due to the quadratic scaling of attention with sequence length.

Research on efficient attention, however, has been driven predominantly by the demands of large language models (LLMs). Approaches such as FlashAttention (Dao et al., 2022; Dao, 2024; Shah et al., 2024; Zadouri et al., 2026), target long, one-dimensional sequences — often with causal masks for autoregressive decoding — and optimize memory access patterns accordingly. Tabular data, on the other hand, differs from language data in ways that directly affect attention performance. First, the sequence lengths for row and column attention are typically asymmetric: the number of columns or features $C$ is usually small (tens to low hundreds), while the number of rows $R$ can range from hundreds to tens of thousands, limited mostly by GPU memory constraints. Therefore, both are much smaller than sequences processed in LLMs. Second, the hidden dimension of current tabular models is relatively small (often in the low hundreds) compared to LLMs, which have hidden dimensions in the thousands, using additional tweaks such as grouped query attention. Third, tabular attention is bidirectional and non-causal, unlike decoder-only LLM architectures. And lastly, the standard five-dimensional tabular tensor layout $(B, R, C, H, D)$ — where $B$ is the batch size, $H$ the number of heads, and $D$ the head dimension — means that column attention operates along a contiguous memory dimension, while row attention requires transposed views with either an explicit `.contiguous()` copy or a backend that supports strided tensors.

In this work, we provide a systematic characterization and benchmark of attention implementations for tabular data. Our key contributions are: (1) We identify the unique characteristics of tabular data and their implications for attention performance. (2) We benchmark five attention backends across a range of realistic shapes for row and column attention. (3) We analyze the results and provide actionable insights for practitioners building tabular foundation models. (4) We release our benchmark code[1] to facilitate future research and the development of attention implementations tailored to the unique characteristics of tabular data.

[1]SAP SE, Germany. Correspondence to: Maximilian Schambach <maximilian.schambach@sap.com>.

*Proceedings of the 2nd ICML Workshop on Foundation Models for Structured Data*, Seoul, South Korea. 2026. Copyright 2026 by the author(s).

---

[1]github.com/SAP-samples/tabular-attention-benchmark

## 2. Background

**General attention:** At its core, the scaled dot-product attention mechanism (Vaswani et al., 2017) computes, for query, key, and value matrices $\mathbf{Q}, \mathbf{K}, \mathbf{V} \in \mathbb{R}^{N \times D}$,

$$\mathrm{Attention}(\mathbf{Q}, \mathbf{K}, \mathbf{V}) = \mathrm{softmax}\left(\mathbf{Q}\mathbf{K}^{\mathrm{T}}/\sqrt{D}\right)\mathbf{V}. \quad (1)$$

It is well known that the naive implementation requires $\mathcal{O}(N^2)$ time and memory. However, several optimized implementations are available that reduce the memory footprint to $\mathcal{O}(N)$. PyTorch's SDPA exposes multiple such backends: the xFormers-based *memory-efficient* and the *cuDNN* backend, which dispatches to NVIDIA's cuDNN library. Furthermore, the FlashAttention (FA) family of implementations (Dao et al., 2022; Dao, 2024; Shah et al., 2024; Zadouri et al., 2026) provides highly optimized attention kernels for NVIDIA GPUs. Whereas FA2 is optimized for Ampere, FA3 is targeting Hopper GPUs and the recent FA4 implementation is primarily tuned for the Blackwell architecture. Beyond the FlashAttention family, SageAttention (Zhang et al., 2025c;a;b) provides an approximate attention implementation, and the vLLM project (Kwon et al., 2023) includes Triton-based implementations. However, in the following we will focus on backends that support both forward and backward passes, which excludes these two inference-only backends.

**Tabular attention:** There are multiple ways to represent tabular embeddings for transformer-based models. With a batch of $B$ tables, with $R$ rows (samples), $C$ columns (features), and $H$ heads of dimension $D$, the standard layout is a five-dimensional tensor of shape $(B, R, C, H, D)$, or equivalently $(B, R, C, d)$ with hidden dimension $d = H \cdot D$, which we refer to as the *row-first* layout. In most recent models, attention is applied alternately along the row and column dimensions: *Column attention* attends across features for each sample, and *row attention* attends across samples for each feature. Hence, for each attention operation, the sequence length is either $C$ or $R$, and the effective batch size is $B \cdot R$ or $B \cdot C$, respectively. The tensor has to be reshaped accordingly to apply the attention operation along the correct dimension, depending on the backend used. The PyTorch-native backends require an input shape of $(\cdots, H, L, D)$ with sequence length $L$, requiring non-contiguous reshapes for both row and column attention. That is, for column attention, the tensor is reshaped to $(B \cdot R, H, C, D)$, which can be represented as a zero-copy view, only requiring a single copy-inducing `.contiguous()` call for the output. The row attention, on the other hand, requires a reshape to $(B \cdot C, H, R, D)$, which cannot be performed using a zero-copy view, needing a `.contiguous()` call for each of the keys, queries, and values. Depending on the size of the tensor, this can lead to significant overhead, as we will observe in the results. This attention pattern holds for all

TabPFNv2-like models but not directly for TabICL which uses a dedicated two-stage attention approach.

Other backends, notably the FA family, require a more compact stride layout with innermost $H$ and $D$ axes. That is, for column attention the tensor can natively be reshaped to $(B \cdot R, C, H, D)$ which is a zero-copy view of the original tensor. In addition, even the row-attention reshape $(B \cdot C, R, H, D)$ can be performed as a zero-copy transpose view and does not require an explicit copy. Hence, it only involves one final `.contiguous()` call to restore a contiguous layout after one iteration of row and column attention for subsequent operations, much fewer memory operations than required by the SDPA backends.

In either case, an asymmetry between column and row attention still exists: Column attention operates along a contiguous memory dimension with shorter sequences, while row attention operates along a non-contiguous dimension with longer sequences. The same argument holds vice versa if using a column-first representation of shape $(B, C, R, d)$, which would make row attention contiguous and column attention non-contiguous. While both layouts incur identical copy overhead, the row-first layout may still be preferred for the overall model: consecutive features of each sample are stored contiguously, which improves cache locality for non-attention operations such as linear or normalization layers. Hence, in the following, we will focus on the row-first layout, which is also the standard layout used in many current tabular models.

Finally, note that the hidden dimension of current tabular models is much smaller than that of current LLMs. This can affect the performance of different backends, and some backends impose restrictions on the number of dimensions they can operate on. For example, FA4 only supports a head dimension of 128 for Hopper GPUs and 64 or 128 for Blackwell GPUs, which is beyond what is typically used in tabular models: TabPFN uses six heads at $D=32$, TabICL uses eight heads at $D=16$, whereas ConTextTab uses twelve heads at $D=64$.

## 3. Experiments

**Tensor shapes:** We benchmark both column and row attention using the row-first tensor layout $(B, R, C, H, D)$. We focus on the case with batch size $B=1$, which allows us to benchmark a wide range of realistic shapes without being limited by GPU memory. Note that we still use the SDPA implementation with `.contiguous()` calls for the row attention even though it is strictly speaking not required for $B=1$, to have a fair comparison that also scales to larger batch sizes in training or inference. For *column attention*, we fix $R=1024$ rows and vary the number of columns $C \in \{16, 32, \ldots, 2048\}$, yielding an effective batch size of $B \cdot R = 1024$ and sequence length equal

to $C$. For *row attention*, we fix $C=64$ columns and vary $R\in\{32, 64, \ldots, 16\,384\}$, giving an effective batch size of $B\cdot C=64$ and sequence length equal to $R$. These ranges reflect realistic tabular datasets, where feature counts are typically moderate and sample counts vary widely. We focus on $H=12$ heads with head dimension $D=64$, a common configuration in tabular transformers (Spinaci et al., 2025) but we also provide results for other configurations – ranging from hidden dimensions 128 to 2048 – in the appendix.

**Backends:** We evaluate five attention backends: (1) SDPA (efficient), PyTorch's xFormers-based backend; (2) SDPA (cuDNN), using NVIDIA's cuDNN library; (3) FlashAttention-2, optimized for Ampere GPUs; (4) FlashAttention-3, which supports strided tensors and targets Hopper GPUs; and (5) FlashAttention-4, tailored to Blackwell GPUs but imposing a minimum head dimension of 128 for Hopper GPUs. All implementation wrappers are tested against the naive `math` PyTorch backend and benchmarked for both the forward and backward pass.

**Hardware and software:** All experiments are conducted on a single NVIDIA H100 NVL GPU with 80 GB memory. We use the latest supported PyTorch and CUDA versions for each backend (PyTorch 2.10.0 with CUDA 13.0 and cuDNN 9.15.1 for SDPA, FA3, and FA4, falling back to PyTorch 2.8 with CUDA 12.8 for FA2), and build FA extensions from source if required. All tensors use `bfloat16` precision, and attention is non-causal throughout.

**Measurement protocol:** The measuring protocol follows closely the implementation in the FlashAttention repository[2]. Each configuration is measured over 50 repetitions preceded by five warmup iterations before each timing sequence. We report throughput in TFLOPS, computed as $4\cdot B_{\text{eff}}\cdot H\cdot L\cdot L\cdot D\,/\,t$ for the forward pass and $2.5\times$ the forward FLOPs for the backward pass. The GPU is warmed up with matrix multiplications before each benchmark run. We measure and report the mean and $\pm 1\sigma$ across repetitions.

## 4. Results

Figure 1 shows the forward and backward throughput of all investigated backends for both column and row attention.

**Column attention:** As only a single `.contiguous()` call is required and the attention operates on the native view, we mostly observe the expected native backend scaling with the sequence length: SDPA (efficient) dominates the very small sequence lengths, likely due to a minimal kernel launch overhead, whereas SDPA (cuDNN) achieves the highest forward throughput for sequence lengths between $C=256$ and 1024 columns. Between $C=16$ and 64, SDPA (efficient) is more than $2\times$ faster than FA3, however

---

[2]https://github.com/Dao-AILab/flash-attention

in absolute throughput differences, the gap is comparably small. In the regime between $C=128$ and 2048, cuDNN dominates, with speedups of between $2.5\times$ and $3\times$ over the SDPA baseline and around $1.5\times$ over FA3 with diminishing returns as the sequence length increases. For longer sequence lengths, FA3 outperforms cuDNN, likely due to more efficient tiling and memory access patterns of its kernel, which is optimized for long-sequence attention. In the backward pass, the SDPA again dominates the very short sequence regime, while cuDNN and FA3 perform on par. Throughout, FA2 shows sub-par performance, likely due to its optimization for the Ampere architecture. On the used H100 GPU, FA4 fails entirely for column attention at $D=64$ due to a minimum M-mode constraint in its kernels, which requires a head dimension of at least 128 on Hopper.

**Row attention:** The picture changes substantially for row attention. FA3 dominates across all sequence lengths except for very short ones below $R=128$, achieving up to $3.5\times$ speedup over the SDPA baseline and around $1.5\times$ speedup over cuDNN. The key driver of this advantage is FA3's input layout convention $(\cdots, L, H, D)$, which preserves the contiguity of the innermost dimensions after the row-attention transpose, requiring fewer memory copies than the SDPA layout $(\cdots, H, L, D)$, as discussed in Section 2, as well as better scaling of the tuned kernel with longer sequences. In the regime between $R=128$ and 4096, which is currently the most relevant for tabular models, this overhead seems to have a significant impact, with diminishing importance as the sequence length increases. Since the memory copy is $\mathcal{O}(N)$ in time, while the attention kernel is $\mathcal{O}(N^2)$, the relative overhead of the copy decreases as the sequence length increases, which is reflected in the diminishing relative gap between FA3 and cuDNN. However, to fully observe this, one would need to go beyond the investigated $R=16384$ limit, which is challenging due to GPU memory constraints on the H100. Note that this crossover behaviour depends on the batch size and hidden dimension, determining the total tensor size and copy time, as we show in the appendix. In the backward pass, FA3 also leads with significant speedups in the most relevant regime.

Further results, using different head dimensions and counts as well as speedup plots, are provided in the appendix, showing patterns consistent with the main results.

**Practical recommendations:** Based on these results, we recommend a mixed strategy for tabular attention in the presented setting (H100 GPU, $H=12$, $D=64$, non-causal attention with `bfloat16` precision): For *column attention* use SDPA (cuDNN) for sequence lengths $C > 64$, and SDPA (efficient) for $C \leq 64$. For *row attention* using FlashAttention-3 across all sequence lengths is feasible, with potential fallback to SDPA (efficient) for very short sequences $R \leq 128$.

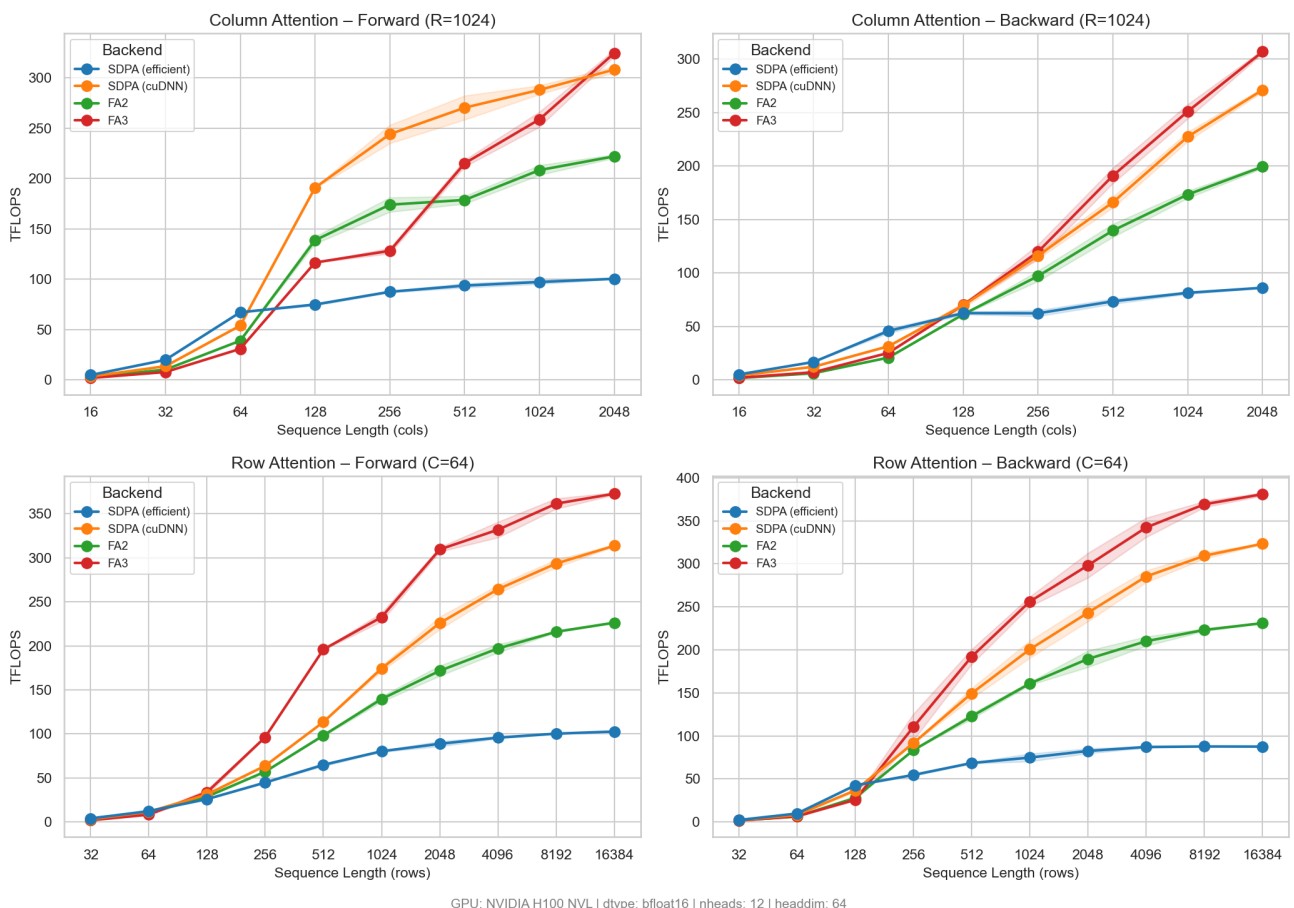

Figure 1. Forward (left) and backward (right) throughput results for column attention (top) and row attention (bottom) for different sequence lengths. Configuration: $H{=}12$ heads, $D{=}64$, `bfloat16`, non-causal. Error bars show $\pm 1\sigma$ over 50 repetitions. Note that FA4 is missing here as it does not support $D{=}64$ on the H100, but additional results are provided in the appendix.

# 5. Conclusion

We have presented a systematic benchmark of attention implementations for tabular foundation models, covering five backends across realistic shapes on an NVIDIA H100 GPU with `bfloat16` precision. We have discussed the unique characteristics of tabular attention, including the asymmetry between row and column attention and the impact of tensor contiguity on performance. Our results show that the optimal backend choice depends critically on the attention pattern: SDPA (efficient) excels at the short column-wise sequences, cuDNN dominates on longer column sequences, while FlashAttention-3 dominates row attention by avoiding costly tensor copies and showing better sequence scaling. The performance gap between the best and worst backend can exceed $3\times$ for a given shape, underscoring the importance of backend selection in tabular deep learning.

A key insight is that the contiguity overhead from `.contiguous()` calls — often dismissed as a minor implementation detail — has a measurable impact on attention

throughput as it dominates the runtime in the medium sequence length regime, depending on the batch size and hidden dimension. Backends that support more compact strides for their input, such as FA3, avoid this overhead to a large extent, providing a consistent advantage. We hope that these findings encourage research and development of custom attention strategies tailored to the unique characteristics of tabular data, such as optimized kernels for the typical shapes and patterns of row and column attention, or novel approaches that can further mitigate the contiguity overhead. We release our benchmark code to facilitate reproducibility and to serve as a testbed for future attention backends targeting this setting.

Our benchmark contains results for the Hopper H100 GPU, which is one of the most widely available GPUs and is commonly used for training tabular models. We leave investigating additional GPU architectures, in particular Blackwell GPUs for which FlashAttention-4 is optimized, as well as investigating inference-only backends, to future work.

## Acknowledgements

We would like to thank Johannes Hoffart, Markus Kohler, and Johannes Höhne for their insightful comments and suggestions throughout the development of this work.

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

# A. Additional Details

## A.1. Quick overview of attention backends

We briefly summarize the key characteristics of the attention backends evaluated in this work. All implementations compute exact scaled dot-product attention but differ in kernel design, hardware specialization, and support for tensor layouts.

**SDPA (efficient):** PyTorch's default memory-efficient attention backend, based on xFormers-style kernels (Lefaudeux et al., 2022). It reduces memory usage via tiling and recomputation, performs well for small to moderate sequence lengths, and has low kernel launch overhead. However, it expects contiguous inputs and does not natively support arbitrary stride layouts.

**SDPA (cuDNN):** A highly optimized backend dispatching to NVIDIA's cuDNN library. It achieves strong performance for a wide range of sequence lengths, particularly in the medium regime, by leveraging vendor-tuned kernels. Similar to the efficient backend, it assumes standard contiguous layouts and may incur overhead from required tensor copies.

**FlashAttention-2 (FA2) (Dao, 2024):** An optimized attention kernel designed for Ampere GPUs, focusing on improved parallelism and work partitioning compared to earlier FlashAttention versions. While memory-efficient and exact, its performance is suboptimal on newer architectures (e.g., Hopper) and it assumes relatively restrictive input layouts.

**FlashAttention-3 (FA3) (Shah et al., 2024):** A further optimized implementation targeting Hopper GPUs, with improved kernel scheduling and support for more flexible stride layouts. In particular, it can operate directly on transposed (non-contiguous) tensors with compact strides, reducing the need for explicit memory copies. This makes it well-suited for row attention in tabular settings.

**FlashAttention-4 (FA4) (Zadouri et al., 2026):** A recent iteration co-designed with newer hardware (e.g., Blackwell GPUs), emphasizing kernel pipelining and asymmetric scaling. FA4 is designed around the observation that tensor core throughput on Blackwell GPUs scales faster than shared memory bandwidth and exponential units, a hardware asymmetry that makes non-matmul operations the bottleneck. FA4 kernels are optimised to minimise shared memory traffic and redundant softmax rescaling. It imposes stricter constraints on input dimensions (e.g., minimum head dimension) and is primarily optimized for large models and longer sequences, making it less applicable in typical tabular configurations.

## A.2. Tensor layout considerations

The standard tensor layout $(B, R, C, H, D)$ places the column dimension before the head dimension. For column attention, the sequence dimension $C$ is already contiguous, so a simple `.view()` suffices. For row attention, the rows and columns must be transposed, producing a non-contiguous tensor.

An alternative layout $(B, C, R, H, D)$ would make row attention contiguous but column attention non-contiguous. Both layouts incur the same theoretical copy cost for their respective non-contiguous direction: the total number of elements copied is identical. However, the standard $(B, R, C, H, D)$ layout may provide better cache locality for the non-attention parts of the model (*e.g.* the MLP layers that process features independently per sample), since consecutive features of the same sample are stored contiguously. Investigating just the `.contiguous()` performance, we were not able to observe any measurable difference, as shown in Table 1.

*Table 1.* Microbenchmark of `.contiguous()` for the row-first layout (Layout A) $(B, R, C, H, D)$ vs. the column-first layout (Layout B) $(B, C, R, H, D)$ with $B{=}2$, $H{=}8$, $D{=}64$. The `transpose(1,2)` call makes the non-contiguous dimension contiguous. Times in μs; bandwidth in GB/s. All ratios are within 0.3% of unity, confirming that layout choice does not affect copy cost.

| | | Time (μs) | | Bandwidth (GB/s) | | |
|---|---|---|---|---|---|---|
| $R$ | $C$ | Layout A | Layout B | Layout A | Layout B | Ratio A/B |
| 1024 | 20 | 65.0 | 64.9 | 1290.6 | 1293.1 | 1.002 |
| 2048 | 50 | 310.3 | 310.8 | 1351.6 | 1349.4 | 0.998 |
| 4096 | 50 | 619.5 | 619.7 | 1354.1 | 1353.6 | 1.000 |
| 8192 | 100 | 2479.5 | 2481.8 | 1353.3 | 1352.0 | 0.999 |
| 8192 | 16 | 396.5 | 397.6 | 1354.0 | 1350.1 | 0.997 |
| 128 | 1024 | 398.8 | 399.3 | 1346.1 | 1344.4 | 0.999 |

### A.3. Contiguity, views, and memory copies in tabular attention

PyTorch tensors are defined by a pointer to storage together with metadata (shape and strides). A tensor is *contiguous* if its elements are laid out in memory such that the stride matches a standard row-major layout. Many high-performance kernels (including SDPA backends) assume or require such a layout.

Operations such as `.view()` or `.reshape()` can return zero-copy views when the requested shape is compatible with the existing stride layout. In contrast, `.transpose()` or `.permute()` typically produce non-contiguous tensors, as they only modify strides without rearranging the underlying storage. For example, a contiguous tensor of shape $(2, 3)$ has strides $(3, 1)$; after `.transpose(0,1)` it has shape $(3, 2)$ but strides $(1, 3)$, meaning elements are no longer laid out contiguously in memory.

The `.contiguous()` call enforces a contiguous layout by allocating new memory and copying the tensor data. This is an $O(N)$ operation in the number of elements and can therefore become a non-negligible overhead, particularly when invoked repeatedly in attention pipelines. As discussed in Section 2, this effect is most pronounced for row attention, where transposing from a row-first layout $(B, R, C, H, D)$ to $(B \cdot C, H, R, D)$ produces a non-contiguous tensor that cannot be represented as a simple view, requiring explicit copies for queries, keys, and values.

In contrast, column attention operates along an already contiguous dimension, allowing reshapes such as $(B, R, C, H, D) \rightarrow (B \cdot R, H, C, D)$ to be implemented as zero-copy views, with only a final `.contiguous()` call needed after the attention operation. Backends that support more flexible stride layouts (e.g., FlashAttention-3) can further reduce the need for intermediate copies by operating directly on transposed views.

In summary, whether an operation incurs a memory copy depends on stride compatibility rather than the operation type itself. In tabular attention, the key distinction is that column attention aligns with the underlying memory layout, while row attention typically does not, making `.contiguous()` calls—and their associated cost—an important practical consideration.

## B. Additional Results

In this appendix, we provide the relative speedup plot and supplementary results for head and dimension configurations beyond the primary $H{=}12$, $D{=}64$ setting discussed in the main text.

### B.1. Hidden dimension and head count variations

We additionally benchmarked $H{=}8$, $D{=}128$; $H{=}16$, $D{=}64$; $H{=}16$, $D{=}128$; and $H{=}32$, $D{=}64$, ranging from the configurations used in TabICL and TabPFN to those used in ConTextTab and beyond. The qualitative findings are consistent with the main results: SDPA (cuDNN) leads for column attention, while FA3 leads for row attention. In those cases where FA4 could be evaluated ($D{=}128$), it performs worse or on par with FA3 for row attention and is not competitive for column attention. This is likely due to the fact that FA4 is mostly optimized for Blackwell GPUs and longer sequences.

**Hidden dimension** 128 **with** 8 **heads at** 16 **(TabICL setup):** The results are shown in Figure 2.

**Hidden dimension** 192 **with** 6 **heads at** 32 **(TabPFN setup):** The results are shown in Figure 3.

**Hidden dimension** 768 **with** 12 **heads at** 64 **(ConTextTab setup):** The results are shown in Figure 4.

**Hidden dimension** 1024 **with** 16 **heads at** 64**:** The results are shown in Figure 5.

**Hidden dimension** 1024 **with** 8 **heads at** 128**:** The results are shown in Figure 6.

**Hidden dimension** 2048 **with** 32 **heads at** 64**:** The results are shown in Figure 7.

**Hidden dimension** 2048 **with** 16 **heads at** 128**:** The results are shown in Figure 8.

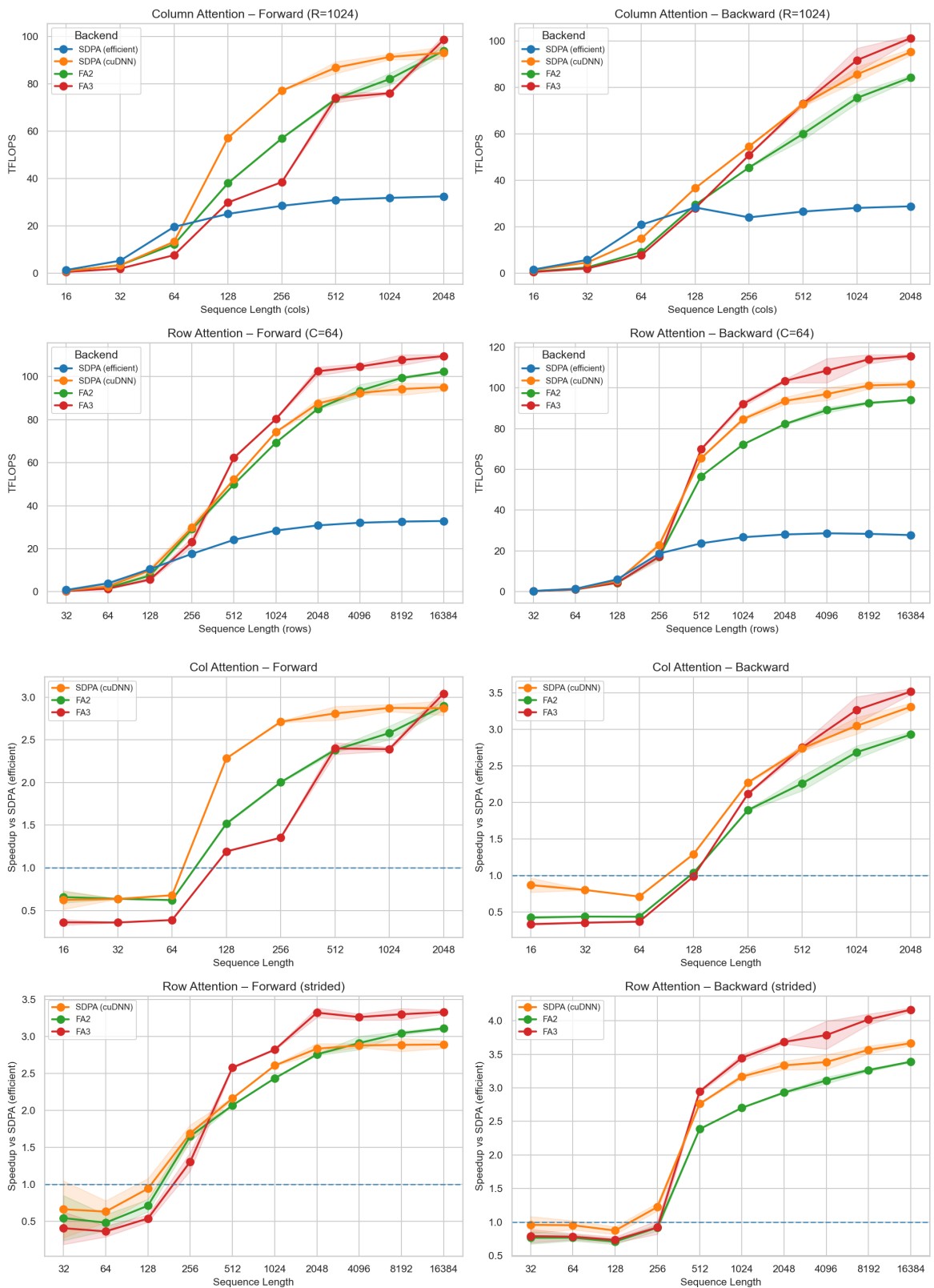

*Figure 2.* Throughput (top) and speedup (bottom) of each backend relative to the SDPA (efficient) baseline in the case of a hidden dimension of $d=128$ consisting of $H=8$ heads with head dimension $D=16$. Error bars show $\pm 1\sigma$ over 50 repetitions.

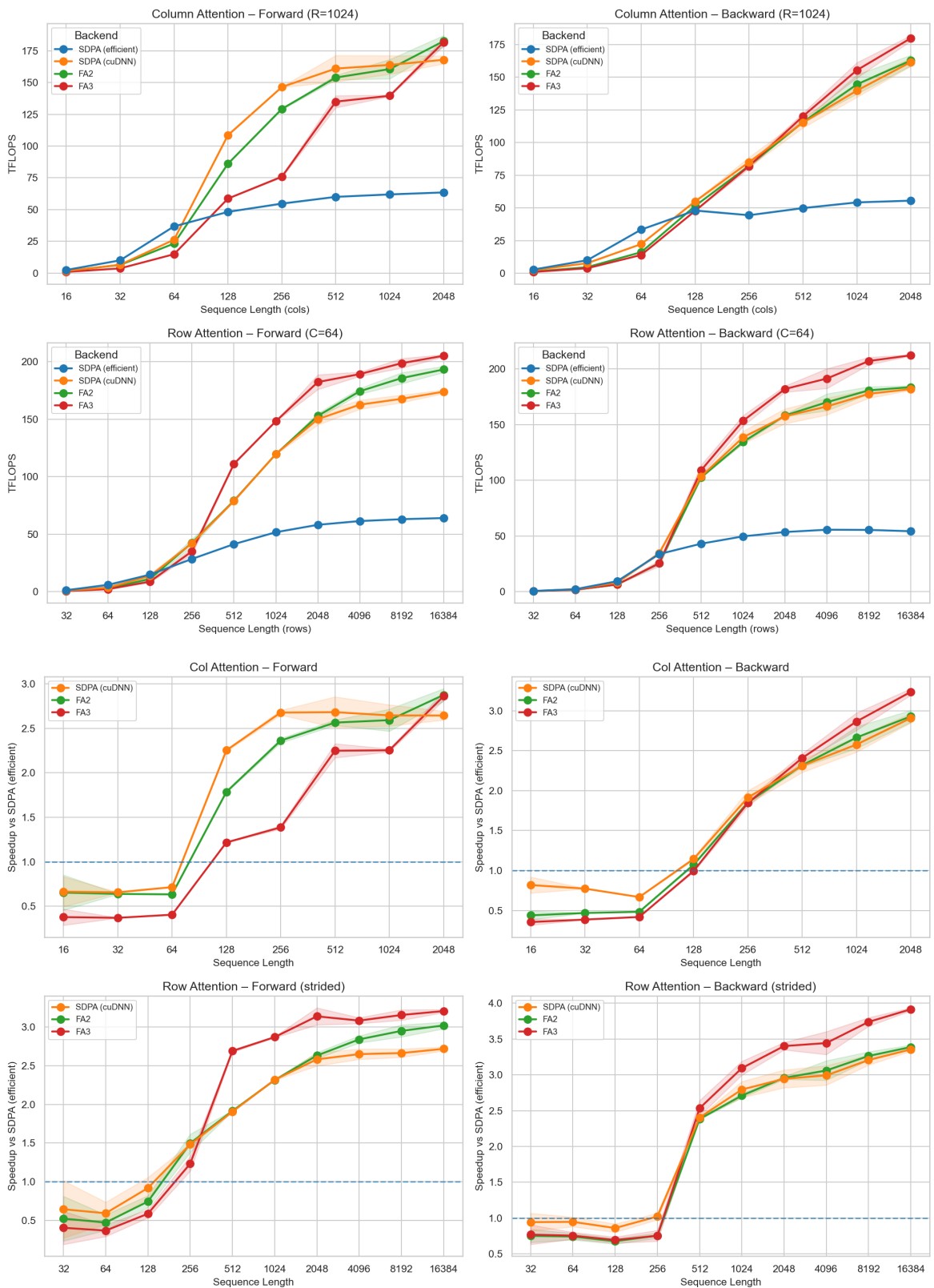

*Figure 3.* Throughput (top) and speedup (bottom) of each backend relative to the SDPA (efficient) baseline in the case of a hidden dimension of $d$=192 consisting of $H$=6 heads with head dimension $D$=32. Error bars show $\pm 1\sigma$ over 50 repetitions.

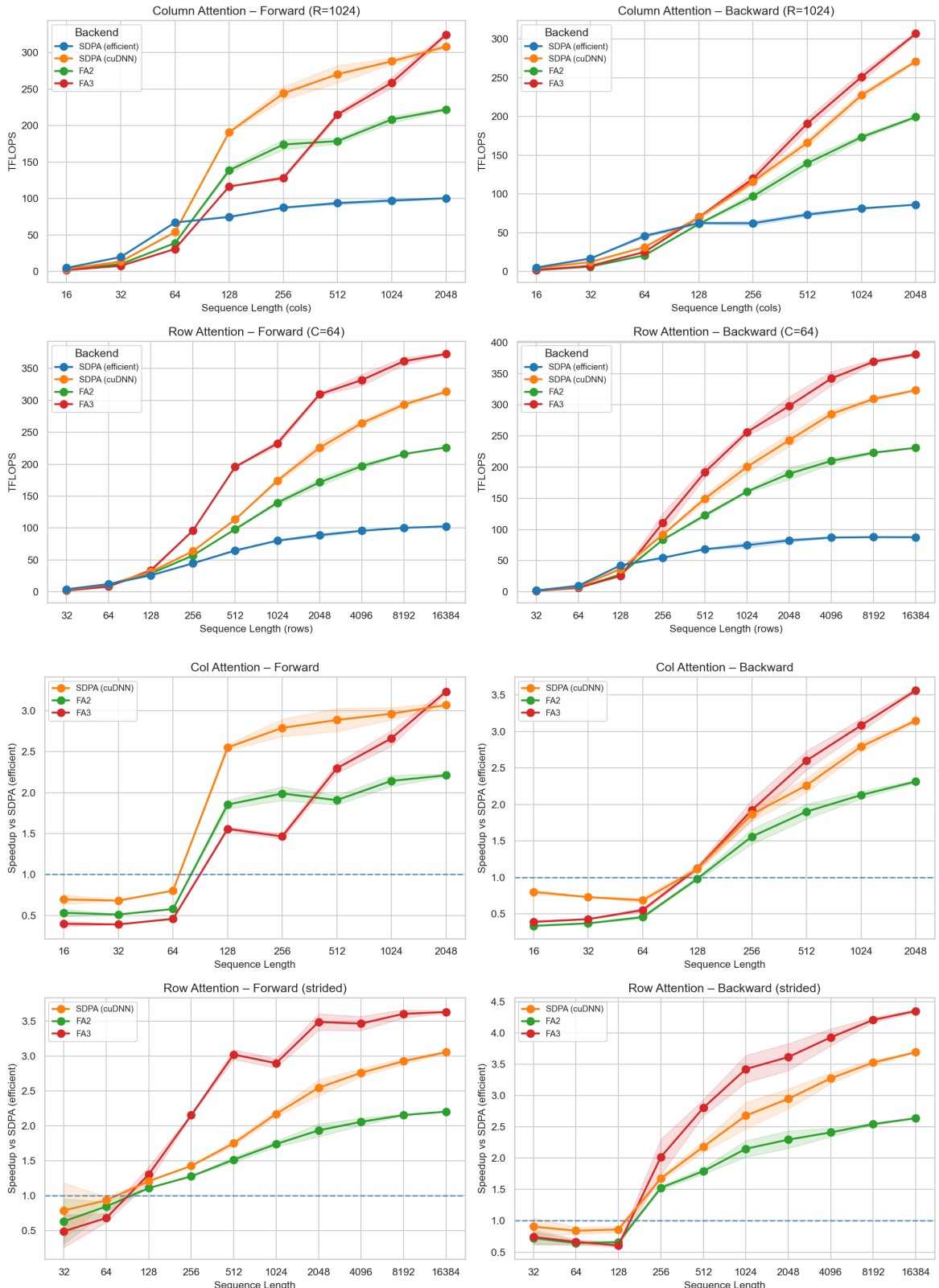

*Figure 4.* Throughput (top) and speedup (bottom) of each backend relative to the SDPA (efficient) baseline in the case of a hidden dimension of $d$=768 consisting of $H$=12 heads with head dimension $D$=64. Error bars show $\pm 1\sigma$ over 50 repetitions.

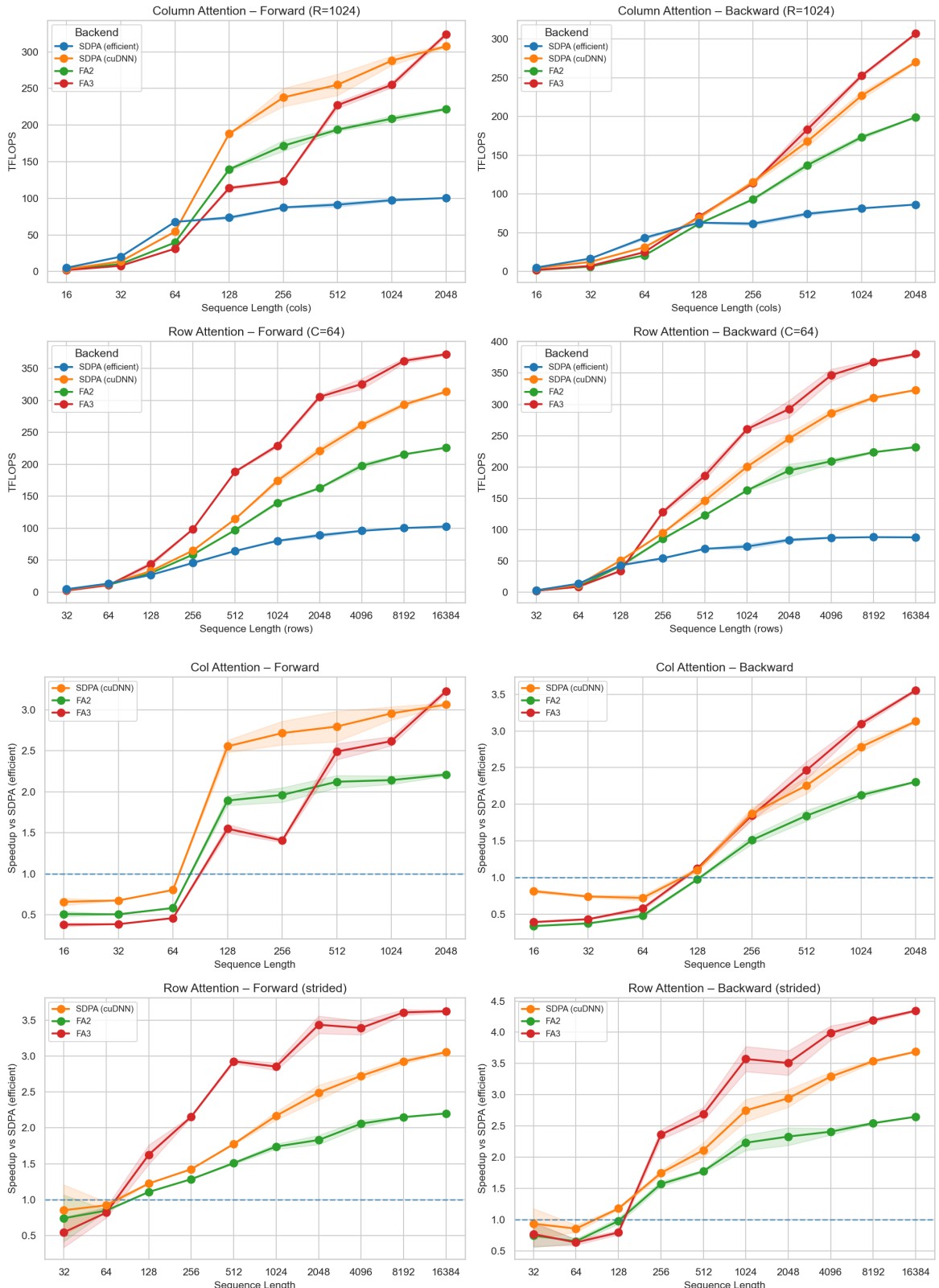

*Figure 5.* Throughput (top) and speedup (bottom) of each backend relative to the SDPA (efficient) baseline in the case of a hidden dimension of $d=1024$ consisting of $H=16$ heads with head dimension $D=64$. Error bars show $\pm 1\sigma$ over 50 repetitions.

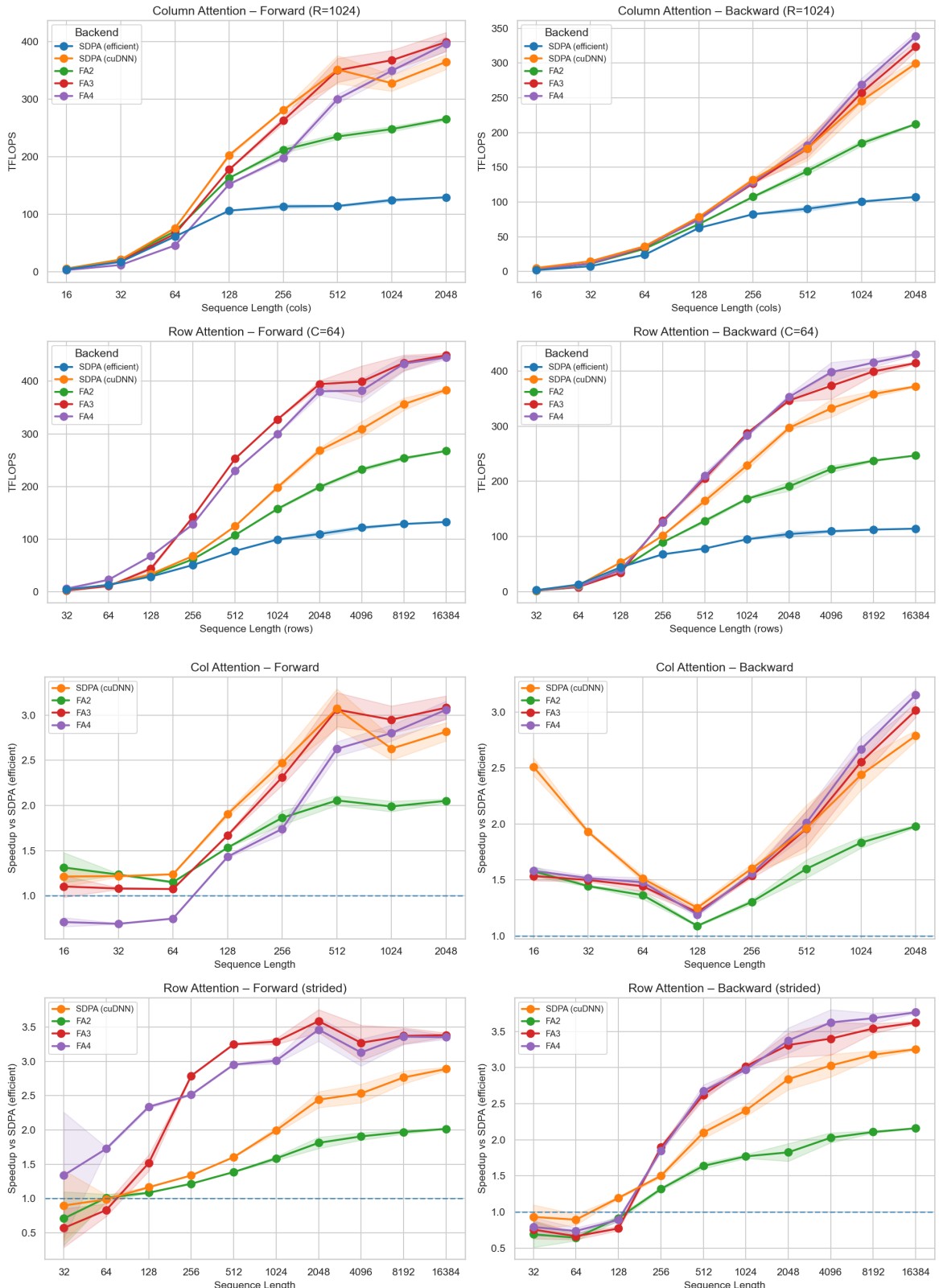

*Figure 6.* Throughput (top) and speedup (bottom) of each backend relative to the SDPA (efficient) baseline in the case of a hidden dimension of $d$=1024 consisting of $H$=8 heads with head dimension $D$=128. Error bars show $\pm 1\sigma$ over 50 repetitions.

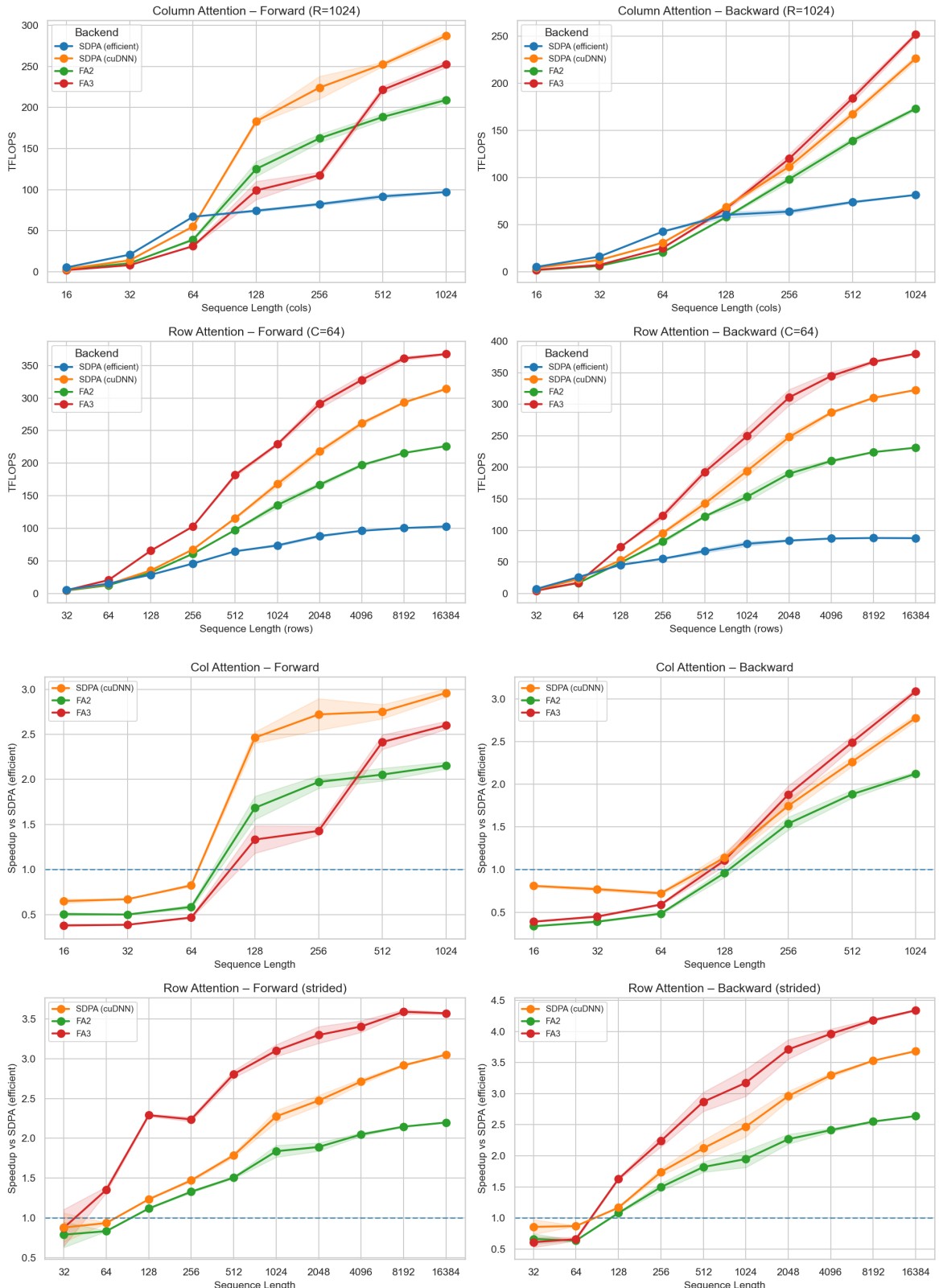

*Figure 7.* Throughput (top) and speedup (bottom) of each backend relative to the SDPA (efficient) baseline in the case of a hidden dimension of $d=2048$ consisting of $H=32$ heads with head dimension $D=64$. Error bars show $\pm 1\sigma$ over 50 repetitions.

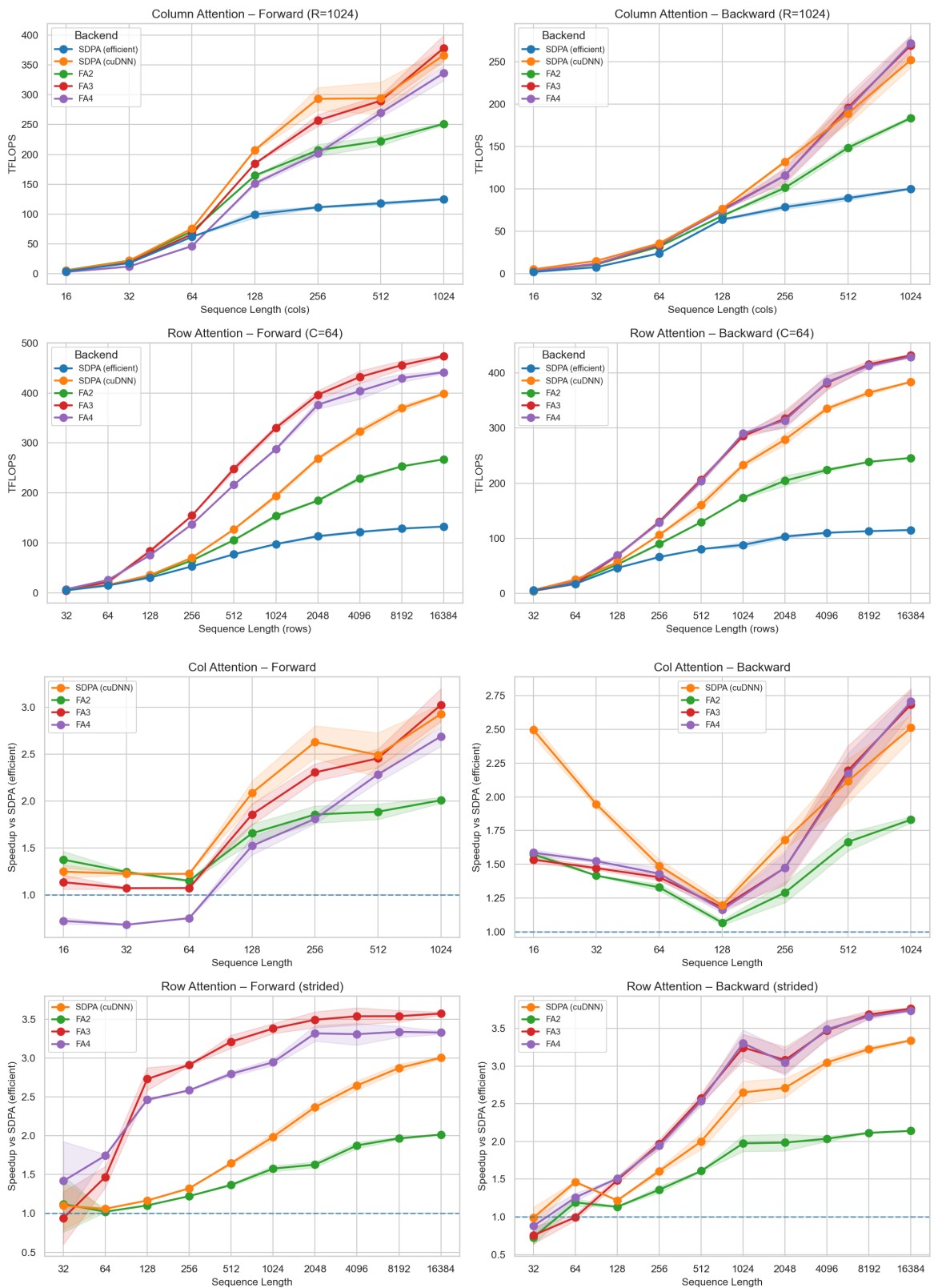

*Figure 8.* Throughput (top) and speedup (bottom) of each backend relative to the SDPA (efficient) baseline in the case of a hidden dimension of $d=2048$ consisting of $H=16$ heads with head dimension $D=128$. Error bars show $\pm 1\sigma$ over 50 repetitions.

