# OpenReview forum: "Benchmarking Attention for Tabular Foundation Models"
_ICML.cc/2026/Workshop/FMSD — FMSD @ ICML 2026 Poster_

### Official Review · Reviewer_jqmT · 2026-05-19
**-**

**Rating:** 6
**Confidence:** 3

**Review:**

Summary
The paper investigates tabular attention characteristics and benchmarks various backends across realistic data shapes on an H100 GPU. It reveals that cuDNN is optimal for column attention at short-to-medium sequence lengths, while FlashAttention-3 dominates row attention.

Strengths:
- Tabular foundation models represent a rapidly growing and high-impact area of research, so highlighting the specific computational bottlenecks of these architectures provides immediate value to the community
- The recommendations section offers clear, hardware-specific guidance (such as using SDPA for small column attention and FA3 for row attention) that practitioners can implement immediately.

Areas of Improvement:
- The exclusive use of an H100 GPU limits the benchmark's broader applicability. While the authors commendably acknowledge this limitation and defer investigating other architectures to future work, evaluating the Ampere-optimized FA2 on an A100 would provide a fairer comparison, and the current inclusion of FA4 feels premature without its intended hardware (Blackwell).
- The study measures the isolated throughput of attention kernels in TFLOPS. Providing a wall-clock time measurement for a full model passes would contextualize the real-world speedup these backend choices actually provide.

Detailed Comments:
- I would try to provide similar experiments on at least one or two different hardware architecture to understand if this results are always specific for a specific hardware or more generelizable

Justification: The paper is a good fit for the workshop and highlights an emerging topic, hardware aware implementation for structured foundation models, which might spark fruitful discussion.

---

### Official Review · Reviewer_u1iB · 2026-05-20

**Rating:** 7
**Confidence:** 3

**Review:**

The paper provides and empirically validates the insight that two different backends should be applied when running attention on rows vs columns. This type of study is relevant due to the amount of compute that can be saved when applying the suggested strategy. The paper mentions a performance gap of 3x between best and worst backend.

When analyzing column attention, I would suggest running the analysis with 4 different R's at least. The paper only contains R=1024. I would say 32, 128, 1024, 4096. Perhaps shifting by a power of 2 if memory allows for it. Similarly, for row attention I would suggest C=16, 64, 128, 512. The paper mostly focuses on ablations regarding number of heads / head dim. This is usually set on the onset and it does not vary as much once the architecture is selected. Now, to avoid a blow-up on number of analysis, for the 4 Rs and 4 Cs that I'm suggested I would recommend only running them on the H=12 and D=64 case.

I would also appreciate an expanded discussion of appendix A.1. I would like to understand at some detail how SDPA is implemented and what the xFormer-style kernes are about.

Also, when discussing O(N) alternatives to attention in lines 63-67 (col 1) I would suggest adding pseudo-code on the appendix that shows why this is the case. I accepted this claim based on my familiarity with FlashAttention.

---

### Official Review · Reviewer_WTKp · 2026-05-21
**Review of "Benchmarking Attention for Tabular Foundation Models"**

**Rating:** 6
**Confidence:** 3

**Review:**

With the rise of tabular foundation models such as TabPFN, there is a lack of low-level operator optimization specifically targeting this architecture. The authors grasped the issue of memory contiguity, the experimental design is reasonable, and the engineering recommendations provided can be directly implemented. It accurately points out that the attention optimization strategies for LLMs cannot be completely applied to tabular data; under standard tensor layouts, column attention is contiguous in memory, but row attention is non-contiguous, requiring expensive transpose and memory copy operations. At the same time, it provides practical conclusions, such as SDPA (especially the cuDNN backend) performing better for column attention, while FlashAttention-3 performs better for row attention.